# Empiric Antifungal Therapy for Intra-Abdominal Post-Surgical Abscesses in Non-ICU Patients

**DOI:** 10.3390/antibiotics12040701

**Published:** 2023-04-03

**Authors:** Eleonora Taddei, Francesca Giovannenze, Emanuela Birocchi, Rita Murri, Lucia Cerolini, Francesco Vladimiro Segala, Pierluigi Del Vecchio, Francesco Taccari, Massimo Fantoni

**Affiliations:** 1Dipartimento di Scienze di Laboratorio e Infettivologiche, Fondazione Policlinico Universitario Agostino Gemelli IRCSS, 00168 Rome, Italy; 2Dipartimento di Sicurezza e Bioetica, Sez. Malattie Infettive, Università Cattolica del Sacro Cuore, 00168 Rome, Italy

**Keywords:** intra-abdominal infections, antifungal therapy, post-surgical infections

## Abstract

The role of empiric antifungals for post-surgical abscesses (PSAs) is controversial, and international guidelines on invasive mycoses focus on bloodstream infections. We analyzed a retrospective cohort of 319 patients with PSA at a tertiary-level hospital in Italy during the years 2013–2018. Factors associated with empiric antifungal administration were analyzed and compared with factors associated with fungal isolation from the abdomen. Forty-six patients (14.4%) received empiric antifungals (65.2% azoles). *Candida* was isolated in 34/319 (10.7%) cases, always with bacteria. Only 11/46 patients receiving empirical antifungals had abdominal *Candida*. Only 11/34 patients with a fungal isolate received empiric antifungal therapy. Upper GI surgery (OR: 4.76 (CI: 1.95–11.65), *p* = 0.001), an intensive care unit stay in the previous 90 days (OR: 5.01 (CI: 1.63–15.33), *p* = 0.005), and reintervention within 30 days (OR: 2.52 (CI: 1.24–5.13), *p* = 0.011) were associated with empiric antifungals in a multivariate analysis, while pancreas/biliary tract surgery was associated with fungal isolation (OR: 2.25 (CI: 1.03–4.91), *p* = 0.042), and lower GI surgery was protective (OR: 0.30 (CI: 0.10–0.89), *p* = 0.029) in a univariate analysis. The criteria for empiric antifungal therapy in our practice seem to be inconsistent with the risk factors for actual fungal isolation. Better guidance for empiric therapy should be provided by wider studies.

## 1. Introduction

Intra-abdominal infections (IAIs) are commonly observed in surgical wards. Among various presentations, post-surgical IAIs are serious healthcare-associated infections (HCAI), complicating 0.5% to 10.1% of abdominal interventions, depending on the type of surgical procedure [1]. A high burden of morbidity and mortality is associated with post-surgical IAIs due to patients’ underlying comorbidities and the high prevalence of multi-drug-resistant organisms (MDRO) [2]. Post-surgical abscesses (PSAs) may be a consequence of delayed surgery for pre-existing uncomplicated IAIs, anastomotic leak, or incomplete source control [3].

Fungal species are more frequently isolated in HCAI than in community-acquired IAIs, with intra-abdominal candidiasis being a manifestation of “deep-seated” or invasive fungal infections. Invasive fungal infections have become increasingly relevant in recent decades and are emergent infections in healthcare settings. In fact, technical advances in intensive care, transplants, and cancer therapy contribute to increase the number of patients that become susceptible to fungal pathogens, with *Candida* spp. being the most common. Invasive candidiasis refers both to bloodstream infections (candidemia) and deep-seated infections (intra-abdominal abscesses, peritonitis, or osteomyelitis) with or without candidemia, the former being the best-studied manifestation [4].

All forms of invasive candidiasis are associated with increased mortality [5]; therefore, prompt recognition and treatment of invasive candidiasis appears to be crucial [6,7]. However, diverse clinical presentations and the infrequent availability of rapid diagnostic assays still represent important challenges. Various treatment strategies for invasive candidiasis have been proposed based on the timing of antimicrobial administration. Prophylaxis is defined as an antifungal regimen administered to asymptomatic patients in the presence of risk factors, while empiric therapy is defined as a treatment based on symptoms suggestive of infection without a microbiological demonstration of a fungal infection. Finally, targeted therapy is defined as an antifungal regimen administered to patients diagnosed with an ascertained fungal infection. Empiric antifungal therapy is then based on the presence of risk factors that are mostly derived from studies of candidemia conducted in the ICU and are then extended to other forms of invasive candidiasis. Classical risk factors include the presence of an indwelling central venous catheter; a history of broad-spectrum antimicrobial therapy; a long-term ICU stay; a recent surgery; necrotizing pancreatitis; medical comorbidities, including diabetes mellitus, cardiac disease, renal failure, and dialysis; total parenteral nutrition; iatrogenic immunosuppression; and multiple sites of colonization with *Candida* spp. [8,9]. In this respect, scoring systems, such as the well-known “*Candida* score”, have been developed [10]. Specific risk factors for *Candida* peritonitis are narrower than those for candidemia and include recurrent gastrointestinal perforations, surgically treated pancreatitis [11], upper gastrointestinal perforations [2,9,12,13], and the previous receipt of antimicrobial therapy [9,14,15,16]. The significance of colonization with *Candida* at multiple body sites as an index of a higher risk of developing intra-abdominal candidiasis is debated [17].

There are several open problems about suspecting intra-abdominal candidiasis: first, there is no consensus on the definition of abdominal candidiasis; second, guidelines propose weak recommendations; third, classical risk factors for invasive candidiasis include abdominal surgery, so virtually all post-surgical patients are at risk; fourth, since candidemia is far more common than other forms of invasive candidiasis, most recommendations are derived from this setting and may not be suited to IAIs.

All these limitations make the choice of empiric antifungal therapy for patients with IAIs very difficult for the prescriber.

The purpose of this work was to analyze the factors associated with the choice of administering empiric antifungal therapy to a retrospective cohort of patients with PSAs, aiming to provide a real-life scenario in a non-ICU setting. We then analyzed and compared risk factors associated with the actual isolation of fungi from the abdomen.

## 2. Results

Three-hundred and nineteen PSA cases were included in our study (males: 55.8%). Two-hundred and nine patients (65.5%) had cancer, and thirty-seven (11.6%) patients received emergency surgery. The population characteristics are detailed in Table 1.

Deep specimen collection was performed for 176 patients out of 319 (55.2%) based on an ultimate decision of a surgeon or interventional radiologist as to the feasibility of the procedure, and intra-abdominal specimens were positive for 154/319 cases (48.3%).

*Candida* spp. was isolated in 34 cases (10.7%) of PSAs. Isolated fungal species included *C. albicans* (17), *C. glabrata* (10), *C. tropicalis* (2), and *C. incospicua* (1). In four cases, more than one *Candida* species was detected. All cases showed bacterial co-infection, mostly by Enterobacteriaceae (22 cases) and Enterococci (15 cases). Only one case of candidemia, caused by *C. tropicalis*, was observed. The characteristics of the subgroup of patients with intra-abdominal *Candida* isolates are also detailed in Table 1. Beta-D-glucan determination was performed for 20 out of 34 cases of *Candida* isolation and was positive in 4 cases (20%).

Based on patients’ records, a QSOFA higher than 2 points at the onset was found for 9 patients (26.5%), and a SOFA score higher than 2 points at the onset was found for 11 patients (32.3%). Most patients with abdominal candidiasis (31/34, 91%) underwent a source control intervention: 13 underwent surgical debridement, 11 underwent percutaneous drainage, 4 underwent percutaneous drainage then surgical debridement, and 3 underwent surgical debridement then percutaneous drainage.

A total of 46 patients out of 319 PSAs (14.4%) received empiric antifungal treatment within 48 h of presentation, with fluconazole being administered in most cases (65.2%). Among those who received empiric antifungal therapy, *Candida* was isolated from the abdomen in 11 cases (23.9%), while in 18 cases abdominal samples were not obtained, 15 patients had negative abdominal cultures, and 2 grew only bacteria. Among 34 patients with fungal isolates, only 11 received empiric antifungal therapy within 48 h of IAI presentation and before culture results. Nine patients did not receive any antifungal treatment during their hospital stay; nevertheless, none of them died. All patients treated with empiric antifungals also received concomitant antibacterial therapy. No adverse events related to antifungal therapy were observed.

We decided to explore which factors influenced the choice of administering empiric antifungals by the ID consultant. Factors associated with empiric antifungal therapy administration for patients with PSAs are reported in Table 2. In a multivariate analysis, upper GI tract surgery, reintervention within 30 days, and an ICU stay in the previous 90 days were independently associated with the prescription of an empiric antifungal therapy for IAIs within 48 h of presentation. Urgent surgery, the presence of an indwelling CVC, and the SOFA score value at IAI presentation were all associated with empiric antifungal therapy in a univariate analysis but were not confirmed in the multivariate analysis.

We then investigated factors associated with fungal isolation from the abdomen in our population in order to compare them with the factors guiding the choice of empiric antifungals. The results of univariate and multivariate analyses for the overall population are reported in Table 3. No risk factors remained significant in the multivariate analysis, but the univariate analysis showed that pancreas/biliary tract surgery was associated with fungal isolation (OR: 2.25 (CI: 1.03–4.91), *p* = 0.042), while a lower GI tract intervention was associated with a lower risk of *Candida* isolation from the abdomen (OR: 0.30 (CI: 0.10–0.89), *p* = 0.029). A trend toward significance was observed for the SOFA value at IAI presentation.

We then repeated the same analysis including only the 177 cases in which intra-abdominal specimens were obtained. We found that lower GI surgery was negatively associated with *Candida* isolation from the abdomen in both the univariate (OR: 0.26 (CI: 0.09–0.78); *p* = 0.017) and multivariate (OR: 0.28 (CI: 0.09–0.86); *p* = 0.027) analyses. Trends toward significant associations of antifungal isolation with previous antibiotic therapy within 30 days (OR: 2.13 (CI: 0.93–4.92); *p* = 0.075) and upper GI surgery (OR: 2.89 (CI: 0.88–9.47); *p* = 0.080) were observed. The results are reported in Table 4.

With regard to outcomes, the crude mortality among the whole PSA population was 3.4%, and a composite adverse outcome of death, no abscess resolution, or abscess recurrence was observed in 15% of cases. For 12 patients with residual abscesses, no other data on recurrence were available after discharge. Among patients growing *Candida* from abdominal specimens (n = 34), 4 patients died (11.8%) and 12 (35.3%) had a composite adverse outcome.

## 3. Discussion

In the present study on patients with PSAs, factors associated with empiric antifungal therapy were analyzed and compared to risk factors for actual fungal isolation from the abdomen.

We found that our prescription practice does not seem to be able to intercept many cases of abdominal candidiasis, even if classical risk factors for invasive candidiasis were mostly taken into account.

Choosing which patients with PSAs are reasonable recipients of empiric antifungal therapy seems to be very relevant due to the high mortality associated with invasive mycoses, but current guidelines and the published evidence are inconclusive.

The Surgical Infection Society guidelines for the management of IAIs [18] consider the problem of empirical coverage for fungi in patients with IAIs. The authors recommend antifungal coverage in severely ill patients when certain risk factors are present (upper gastrointestinal perforations, recurrent bowel perforations, surgically treated pancreatitis, and prolonged courses of broad-spectrum antibiotic therapy) and for those who are known to be heavily colonized with *Candida*. The authors also comment that the isolation of *Candida* spp. and inadequate empiric therapy are associated with higher mortality.

The World Society of Emergency Surgery [19], in its paper on IAI management, recommends empiric antifungal therapy in two cases, patients with septic shock and patients with post-operative IAIs, yet it admitted that data on the latter group of patients are scarce and that the indication for antifungal coverage is inferred from studies showing that *Candida* isolation per se is linked to poor prognoses in critically ill patients [5,20,21,22].

The choice of administering empiric antifungal therapy in our study was based on classical factors, such as upper GI surgery, abdominal reintervention, and indwelling CVCs, as well as hallmarks of severity, including a recent ICU stay, a higher SOFA score, and emergency surgery. We did not take into account multiple-site *Candida* colonization since these data are not routinely available for our patients, and data on total parenteral nutrition were not analyzed due to their unavailability in our database. In our small population, on the other hand, the only factor with a strong negative association with *Candida* isolation from the abdomen was a history of lower GI surgery. The only risk factor for *Candida* isolation, on the other hand, was pancreas or biliary tract surgery, which was not confirmed when only patients with available intra-abdominal samples were analyzed.

As to the choice of empirical therapy, echinocandins are preferred over azoles by Mazuski for “higher risk” patients. Sartelli agrees that echinocandins should probably be used as an empirical antifungal therapy in critically ill patients with CA-IAIs or HA-IAIs, while s first-line fluconazole therapy is preferable in the other cases.

In our daily practice, we tended to prefer an empiric antifungal therapy with azoles for two reasons. First, based on pharmacokinetics, azoles are small polar molecules with better penetration into body tissues, including the peritoneum [23]. Moreover, a wide utilization of echinocandins entails a selective pressure, leading to antifungal resistance, and therefore contrasts with antimicrobial stewardship principles [24].

As for diagnosis, it must be noted that a risk of contamination is inherently present when environmental microorganisms, such as fungi, are isolated in culture. For this reason, we included only positive cultures from freshly drawn abdominal specimens (drainage in place for less than 24 h) or intra-operatory specimens, as we perform in our daily practice. It has been suggested [25] to use non-culture-based methods for the diagnosis of IAC in patients with at least one risk factor for abdominal candidiasis. Our patients mostly were not tested for beta-D-glucan plasma levels due to unavailability at the time of this study. When tested, no correlation was found between beta-D-glucan levels and *Candida* spp. isolation from the abdomen. In fact, beta-D-glucan has been best-studied in patients with candidemia and is more useful for excluding bloodstream fungal infection. Moreover, false-positive results (due to mold contamination and medication gauzes) are possible in surgical patients.

The present study had may limitations. First, due to the retrospective design, some relevant variables were missing. In particular, some data for calculating the *Candida* score were unavailable. Moreover, our sample size was small, and the actual abdominal candidiasis cases were too few to allow for further analysis, including an outcome analysis. Moreover, due to the peculiarities of our population, such as the absence of ICU and hematologic patients and the high rate of patients with cancer, the generalizability of the results is limited. We believe, however, that our observation that the factors associated with empiric antifungal administration for PSAs, which included some classical risk factors for invasive candidiasis, were not consistent with the factors associated with actual fungal IAIs deserves some interest. Our experience with a real-life cohort, with its intrinsic limitations, reminds us that clinical decisions must sometimes be made based on less-than-optimal data. Wider, well-designed studies should provide guidance for empiric antifungal administration to patients with PSAs and analyze its impact on relevant outcomes.

## 4. Materials and Methods

### 4.1. Inclusion Criteria

A retrospective single-center observational study was conducted at our tertiary-level teaching hospital based in Rome, Italy. All adult inpatients with a diagnosis of a post-surgical intra-abdominal abscess (PSA) that were hospitalized during the years 2013–2018 were eligible to be included in this study. At our center, an infectious disease consultation team (IDCT) operates. It performs daily elective ID consultations on demand throughout all hospital wards, including the surgical and medical wards, with the exception of the ICU and hematology. ICU and hematology inpatients were thus excluded since they receive different forms of ID consultation (i.e., urgent ID consultation, as requested, and emerging infections are managed directly by the caring physicians).

### 4.2. Definitions and Data Collection

A PSA was defined as a fluid collection within the peritoneum that presented following an abdominal surgical intervention and was diagnosed using abdominal ultrasound or an abdominal computed tomography (CT) scan. All cases of intra-hepatic abscesses, solid organ infections due to hematogenous spread, and abscesses secondary to inflammatory bowel disease (IBD) were then excluded.

Each patient’s historical electronic record was retrospectively examined, and the data were extracted based on a pre-specified case report form (CRF). The CRF for each patient included their age; sex; ward of admission; relevant comorbidities; history of previous (90 days) ICU stay or MDRO isolation before the PSA diagnosis; previous (30 days) antibiotic use; indication and timing of abdominal surgery; abdominal reintervention within 30 days; number, size, and location of intra-abdominal abscesses; presence of a persistent intra-abdominal leak or fistula; clinical conditions and SOFA score at the time of IAI diagnosis; indwelling CVC at the time of PSA diagnosis; source control procedures; identified abdominal pathogens and results of susceptibility testing; concomitant blood culture results; data on empiric and definitive antimicrobial therapy; length of hospital stay; length of the available follow-up; and outcome information.

Relevant comorbidities included any history of the following: ischemic heart disease (IHD), cerebrovascular disease, peripheral arteriopathy, chronic obstructive pulmonary disease (COPD), diabetes mellitus (DM), chronic kidney disease (CKD, stages III-V), moderate to severe liver disease (Child–Pugh classes B-C), HIV infection, rheumatological disease, dementia, solid or hematologic malignancies, obesity (body mass index (BMI) > 30 kg/m^2^), and malnutrition (BMI < 18 kg/m^2^).

Empiric antifungal therapy was defined as a therapy with echinocandins or fluconazole administered within forty-eight hours of IAI presentation in the absence of any fungal isolate.

Only cultures from freshly drawn percutaneous specimens (drainage in place for less than 24 h) or from intra-abdominal specimens obtained through surgical source control were considered when determining the etiology of an IAI [26].

### 4.3. Outcomes

The clinical outcome was defined as abscess resolution or significant size reduction at imaging with no history of relapse. Radiological evidence of absent resolution or abscess recurrence and a patient’s death were considered together as a composite unfavorable outcome.

### 4.4. Statistics

Descriptive statistics were used to summarize the results. Associations between categorical variables were analyzed using Pearson’s chi-square test or a two-tailed Fisher’s exact test, as appropriate. A logistic regression was used to evaluate the factors associated with empiric antifungal therapy administration for fungal isolation from the abdomen. Odds ratios with 95% confidence intervals (CI) were determined for each factor. Factors associated with empiric antifungal therapy or with fungal isolation from the abdomen at the 0.1 significance level in a univariate analysis were entered in the full multivariable model using a backward stepwise approach. The IBM SPSS Statistics software, version 23 (IBM Corporation, Armonk, NY, USA), was used for the statistical analysis.

## 5. Conclusions

Criteria for the empiric administration of antifungals for IAIs or PSAs are not provided by guidelines, so much is left to the physician’s decision. It is crucial to better establish the risk factors for *Candida* infection of the abdomen. We found that pancreas and biliary tract surgeries were a significant risk factor in our small group of patients, while this was not considered in our daily practice when deciding to start empiric antifungal therapy for PSAs. A wider population and a prospective design are needed to define which patients with PSAs could benefit from prompt antifungal treatment, if there is an advantage from a clinical standpoint, whether the benefits of empiric antifungal therapy outweigh the costs, and the ecological impact of antifungal coverage for surgical patients.

## Figures and Tables

**Table 1 antibiotics-12-00701-t001:** Characteristics of the patient population.

	Total (n = 319)	*Candida* spp. Not Isolated from the Abdomen (n = 285)	*Candida* spp. Isolated from the Abdomen (n = 34)
Age, mean (±SD)	59.81 (15.07)	59 (±15)	64 (±15)
Sex, male (%)	178 (55.8)	157 (55.1)	22 (64.7)
Hemodialysis (%)	3 (0.9)	2 (0.7)	1 (2.9)
Cardiovascular disease (%)	15 (4.7)	14 (4.9)	1 (2.9)
Indwelling CVC at IAI diagnosis (%)	78 (24.5)	67 (23.5)	11 (32.3)
Diabetes (%)	8 (2.5)	7 (2.5)	1 (2.9)
ICU within 90 days (%)	20 (6.3)	16 (5.6)	4 (11.8)
MDRO within 3 months (%)	10 (3.1)	7 (2.5)	3 (8.8)
Antibiotic treatment within 30 days (other than surgical prophylaxis) (%)	69 (21.6)	58 (20.4)	11 (32.4)
Previous CT or RT within 30 days (%)	16 (5.0)	15 (5.3)	1 (2.9)
Cancer diagnosis (%)	209 (65.5)	190 (66.7)	19 (55.9)
SOFA at IAI diagnosis, median (IQR)	0.00 (0.00–1.00)	0 (0–1)	1 (0–2)
Fever at IAI diagnosis (%)	270 (84.6)	239 (83.9)	31 (91.2)
Worsening abdominal pain/abdominal examination/purulent abdominal drainage at IAI diagnosis (%)	141 (44.2)	125 (43.9)	16 (47.1)
Site of surgery (%) - Upper GI tract - Lower GI tract - Pancreas and biliary tract - Genitourinary - More than one of the previous sites ∘ Bricker ∘ Duodenal cephalopancreasectomy - Others	30 (9.4) 91 (28.5) 61 (19.1) 47 (14.7) 80 (25.1) 15 (4.7) 34 (10.7) 10 (3.1)	25 (8.8) 87 (30.5) 50 (17.5) 44 (15.4) 75 (26.3) 11 (3.9) 33 (11.6) 4 (1.4)	5 (14.7) 4 (13.8) 11 (32.4) 3 (8.8) 5 (14.7) 4 (11.8) 1 (2.9) 6 (17.6)
Reason for surgery (%) - Appendicitis, cholecystitis, or diverticulitis - Perforation - Stenosis, occlusion, or ischemic event - Cancer - Obesity - Others	24 (7.5) 13 (4.1) 12 (3.8) 222 (69.6) 16 (5.0) 28 (8.8)	19 (6.7) 11 (3.9) 11 (3.9) 201 (70.5) 15 (5.3) 24 (8.4)	5 (14.7) 2 (5.9) 1 (2.9) 21 (61.8) 1 (2.9) 4 (11.8)
Persistent leak or fistula (%)	53 (16.6)	47 (16.5)	6 (17.6)
Urgent surgery (%)	37 (11.6)	30 (10.5)	7 (20.6)
Second surgery in the following 30 days (%)	119 (37.3)	103 (36.1)	16 (47.1)
BSI secondary to IAI (%)	47 (14.7)	42 (14.7)	*Candida* spp. and bacteria: 5
Deep specimen collection (%) - not performed - positive culture - negative culture	143 (44.8) 154 (48.3) 22 (6.9)	120 (42.1)	34 (100)
Polymicrobial abdominal isolate (%)	122 (38.2)	88 (30.9)	34 (100)
Fungal growth in abdominal culture (%) - *Candida* albicans (%) - *Candida* glabrata (%) - *Candida* tropicalis (%) - Other *Candida* spp. (%) More than one *Candida* spp. (%)	34 (10.7)		34 17 (50) 10 (29.4) 2 (5.9) 1 (2.9) 4 (11.8)
Abdominal collection diameter > 3 cm (%)	257 (80.6)	231 (81)	26 (76.5)
More than one abdominal collection (%)	169 (53.0)	151 (53)	18 (52.9)
Definitive source control obtained (%)	212 (66.4)	181 (63.5)	31 (82.4)
Type of source control (%) - Surgical - Percutaneous - Percutaneous then surgical - Surgical then percutaneous	70 (21.9) 110 (34.5) 23 (7.2) 9 (2.8)	57 (20) 99 (34.7) 19 (6.7) 6 (2.1)	13 (50) 11 (47.1) 4 (8.8) 3 (8.8)
Time to adequate source control, median (IQR)	4 (1–10)	3 (1–8)	7 (2–20)
Antifungal at any time during therapy (%)	86 (27.0)	63 (22.1)	23 (67.6)
Antimicrobial treatment duration, median (IQR)	18 (12–26)	18 (12–25)	24 (14–32)
Treatment duration after source control, median (IQR)	15.5 (8.25–23.75)	16 (9–23)	15 (8–28)
Antibiotic therapy for IAI longer than 14 days (%)	205 (64.3)	182 (63.9)	21 (61.8)
Unfavorable outcome (relapse or death) (%) - Relapse - Death	46 (14.4) 35 (11) 11 (3.4)	34 (11.9) 27 (9.5) 7 (2.5)	12 (35.3) 8 (23.5) 4 (11.8)
LOS, median (IQR)	17 (11–28)	16 (11–27)	49 (25–74)
Length of FU, median (IQR)	221 (41–637)	228 (39–649)	102 (51–466)
Empiric antifungal therapy Azoles Echinocandins Azoles then Echinocandins Echinocandins then Azoles	46 (14.4)	35 (12.3)	11 (32.4) 21 (61.8) 13 (18.2) 2 (5.9) 1 (2.9)

Abbreviations: BSI: bloodstream infection, CT: chemotherapy, FU: follow-up, GI: gastrointestinal, IAI: intra-abdominal infection, ICU: intensive care unit, IQR: inter-quartile range, LOS: length of stay, MDRO: multi-drug-resistant organism, RT: radiotherapy, SD: standard deviation, SOFA: sequential organ failure assessment.

**Table 2 antibiotics-12-00701-t002:** Factors associated with the administration of empiric antifungal therapy.

	Univariate		Multivariate	
	OR (95% CI)	*p*	OR (95% CI)	*p*
ATB in previous 30 days	1.53 (0.75–3.09)	0.240		
Urgent surgery	3.74 (1.75–7.99)	0.001	1.88 (0.74–4.80)	0.185
Chemotherapy	1.40 (0.38–5.10)	0.614		
Neoplasia	0.64 (0.34–1.21)	0.168		
Upper GI surgery	4.20 (1.85–9.56)	0.001	4.76 (1.95–11.65)	0.001
Pancreas/biliary tract surgery	1.03 (0.47–2.27)	0.934		
Lower GI surgery	0.66 (0.31–1.39)	0.273		
Reintervention within 30 days	2.95 (1.55–5.64)	0.001	2.52 (1.24–5.13)	0.011
CVC	2.28 (1.18–4.41)	0.014	1.05 (0.46–2.38)	0.914
ICU in previous 90 days	7.31 (2.84–18.76)	<0.0001	5.01 (1.63–15.33)	0.005
Fever at IAI presentation	1.01 (0.42–2.42)	0.977		
SOFA score at IAI presentation	1.25 (1.05–1.48)	0.012	1.16 (0.96–1.41)	0.122

Abbreviations: ATB: antibiotics, CVC: central venous catheter, GI: gastrointestinal, IAI: intra-abdominal infection, ICU: intensive care unit, SOFA: sequential organ failure assessment.

**Table 3 antibiotics-12-00701-t003:** Factors associated with *Candida* isolation from the abdomen in the overall population (n = 319).

	Univariate		Multivariate	
	OR (95% CI)	*p*	OR (95% CI)	*p*
ATB in previous 30 days	1.87 (0.86–4.06)	0.112		
Urgent surgery	2.05 (0.83–5.09)	0.122		
Chemotherapy	0.55 (0.07–4.26)	0.563		
Neoplasia	0.63 (0.31–1.30)	0.214		
Upper GI surgery	1.79 (0.64–5.04)	0.268		
Pancreas/biliary tract surgery	2.25 (1.03–4.91)	0.042	1.70 (0.74–3.89)	0.210
Lower GI surgery	0.30 (0.10–0.89)	0.029	0.39 (0.13–1.20)	0.101
Reintervention within 30 days	1.56 (0.76–3.19)	0.222		
CVC	1.56 (0.72–3.36)	0.260		
ICU in previous 90 days	2.24 (0.70–7.14)	0.172		
Fever at IAI presentation	1.99 (0.58–6.78)	0.272		
SOFA score at IAI presentation	1.17 (0.97–1.41)	0.091	1.12 (093–1.35)	0.249

Abbreviations: ATB: antibiotics, CVC: central venous catheter, GI: gastrointestinal, IAI: intra-abdominal infection, ICU: intensive care unit, SOFA: sequential organ failure assessment.

**Table 4 antibiotics-12-00701-t004:** Factors associated with *Candida* isolation from the abdomen when an intra-abdominal sample was available (n = 176).

	Univariate		Multivariate	
	OR (95% CI)	*p*	OR (95% CI)	*p*
ATB in previous 30 days	2.13 (0.93–4.92)	0.075	2.30 (0.97–5.47)	0.059
Urgent surgery	1.74 (0.66–4.58)	0.266		
Chemotherapy	0.36 (0.04–2.89)	0.337		
Neoplasia	0.69 (0.32–1.47)	0.335		
Upper GI surgery	2.89 (0.88–9.47)	0.080	2.32 (0.68–7.88)	0.177
Pancreas/biliary tract surgery	1.86 (0.82–4.26)	0.140		
Lower GI surgery	0.26 (0.09–0.78)	0.017	0.28 (0.09–0.86)	0.027
Reintervention within 30 days	1.05 (0.5–2.23)	0.893		
CVC	1.86 (0.82–4.26)	0.140		
ICU in previous 90 days	3.02 (0.80–11.38)	0.102		
Fever at IAI presentation	2.32 (0.66–8.16)	0.191		
SOFA score at IAI presentation	1.09 (0.90–1.31)	0.389		

Abbreviations: ATB: antibiotics, CVC: central venous catheter, GI: gastrointestinal, IAI: intra-abdominal infection, ICU: intensive care unit, SOFA: sequential organ failure assessment.

## Data Availability

The full dataset and statistical code are available from the corresponding. Author and managed according to local ethical policies.

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
