# Peer review of "Empiric Antifungal Therapy for Intra-Abdominal Post-Surgical Abscesses in Non-ICU Patients"

_antibiotics, 2023, doi:10.3390/antibiotics12040701_

Round 1

Reviewer 1 Report

1- please make the abstract structured based on MDPI style

2- you need to try to shorten the introduction section, specifically, I don't like the way you have many small paragraphs, you should merge them together 

3- make subsections in the method section 

Author Response

Thank you very much for your comments, please find our reponse below:

1- please make the abstract structured based on MDPI style

We structured the abstract based on Instructions for Authors in Antibiotics, stating that “The abstract should be a total of about 200 words maximum. The abstract should be a single paragraph and should follow the style of structured abstracts, but without headings…”. We reduced words count.

2- you need to try to shorten the introduction section, specifically, I don't like the way you have many small paragraphs, you should merge them together 

We revised some sections of the Introduction according to your helpful suggestions.

3- make subsections in the method section 

We re-formatted the Methods section according to your suggestions.

Reviewer 2 Report

The present manuscript is on the so interesting topic of Candida infection in intra-abdominal abscesses (IAA). The study is retrospective with its inherent limitations though it presenta  a relatively large cohort of patients with IAA, half of them with specimen collection. Authors try to find the association of antifungal empirical therapy with outcomes. The study is interesting though I have many comments:

-                     A direct table comparing those patients with and without Candida isolation must be provided. Table 1 shows all cohort and Candida patients. I would suggest to include one column (or additional table) showing “entire cohort”, “no candida”, “candid” to better see differences.

-                     Many important objective variables (dialysis, Charlson comorbidity or other comorbidities like diabetes or hypertension, days on parenteral nutrition) are lacking while other temporal changing are reported (fever, abdominal pain, etc). Laboratory data when abscesses are diagnosed should be provided (renal, leucocytes, RCP, Procalcitonin if available, etc.) to assess their association with Candida isolation.

-                     An important point is that the analysis should strictly be done in those with deep specimen sampling: in those who are not punctioned we do not really know if they had Candida. This is an important bias. Authors must try to clarify why one patients were punctioned while others were and others not: size? Location? It is mandatory to also make this sub-analysis since we can assume that many of the non-punctioned patients could indeed have Candida in these IAA; in fact authors state that even 9 patients patients with proven Candida did well without specific antifungal treatment. It challenges the conclusions that empirical treatment improves prognosis. I am not so clear when looking that many of the isolated species could be resistant to azoles.

-                     Analyzed endpoints are debatable. It is logical to create a combined endpoint when looking at relatively low mortality, but probably the population is not sick enough to only look for a hard endpoint (death).

-                     I do not agree with using azoles as first-line antifungal therapy when about a quarter of patients have resistant species. It is not contrary to a stewardship program if descalation is promptly done after non-isolation or isolation of an azol-sensitive species. How many of these antifungal therapies were really active against the isolated species?

-                     Many factors related with worse outcomes are mainly explicatives since occurring concomitantly to patients evolution: the larger the number of days the worse the outcomes is only reflecting a patient in whom antibiotics are maintained for a longer period since he is doing bad.

-                     There is a relatively small number of patients with Candida isolation. This is a major limiting factor when trying to perform a retrospective multivariable analysis in such difficult cohort (patients with large admissions and lots of uncontrolled biases and neglected variables). I am not sure if Candida really plays a major pathogen role or is merely  reflexing the disease severity; looking at very low positive rate of B-D-glucan and that all Candida isolation were polymicrobial one would think so...

-                     It must taken into account that Candida can be difficult to isolate in cultures and maybe we have many false negatives among those patients receiving antifungal therapy. It is logical to think that many of them were negative because of antifugals. In this sense, molecular techniques probably would add t the diagnosis.

-                     I suggest authors to provide and assess the prognostic role of other isolated bacteria in the samples analyzed. It would be interesting to see the role of the type of isolated bacteria (BGN, entorococci, polymicrobial).

My overall impression after reading the manuscript is that there are many few  Candida isolations, there is a large proportion of patients in whom abscesses are not analyzed and there a lot of uncontrolled factors and incoherencies along the manuscript to finally state that empirical antifungal therapy improves outcomes. Manuscript should be completely redone to sound more convincing with these conclusions.

Author Response

Thank you very much for your punctual observations, which made us restructure our paper. Please find our responses below:

-                     A direct table comparing those patients with and without Candida isolation must be provided. Table 1 shows all cohort and Candida patients. I would suggest to include one column (or additional table) showing “entire cohort”, “no candida”, “candid” to better see differences.

               We reformatted Table 1 according to your suggestion.

-                     Many important objective variables (dialysis, Charlson comorbidity or other comorbidities like diabetes or hypertension, days on parenteral nutrition) are lacking while other temporal changing are reported (fever, abdominal pain, etc). Laboratory data when abscesses are diagnosed should be provided (renal, leucocytes, RCP, Procalcitonin if available, etc.) to assess their association with Candida isolation.

We were unable to retrieve all risk factors for Candida infection, but as you suggested, we added some variables that available in our database: Cardiovascular Disease, Chronic Kidney Disease, Dialysis (many data missing) , Indwelling CVC at the time of IAI diagnosis. Data on total parenteral nutrition and laboratory data are not available unfortunately. 

  •                     An important point is that the analysis should strictly be done in those with deep specimen sampling: in those who are not punctioned we do not really know if they had Candida. This is an important bias.

We eliminated the outcome analysis, as suggested below. 

Authors must try to clarify why one patients were punctioned while others were and others not: size? Location?

As it is frequent in clinical practice, we believe, drainage of abdominal collections is sometimes not performed due to anticipated complications (e.g. abscess location) or abscess dimension. We believe that it is interesting to appraise that in a real-life cohort, even in a tertiary level hospital with an Interventional Radiology Team available, only about two thirds of patients undergo source control.

It is mandatory to also make this sub-analysis since we can assume that many of the non-punctioned patients could indeed have Candida in these IAA; in fact authors state that even 9 patients with proven Candida did well without specific antifungal treatment. It challenges the conclusions that empirical treatment improves prognosis. I am not so clear when looking that many of the isolated species could be resistant to azoles.

We appreciate this observation about patients showing a good outcome even without direct antifungal therapy, and decided to eliminate any outcome analysis accordingly, focusing on the choice of empiric therapy and risk factors for Candida isolation.

-                     Analyzed endpoints are debatable. It is logical to create a combined endpoint when looking at relatively low mortality, but probably the population is not sick enough to only look for a hard endpoint (death).

A composite outcome was chosen due to small numbers.

However, as reported further in response to your observations, we agree to focus this work on a discussion of empirical antifungal prescription rather than outcome analysis, due to small numbers and the lack of key variables.

-                     I do not agree with using azoles as first-line antifungal therapy when about a quarter of patients have resistant species. It is not contrary to a stewardship program if descalation is promptly done after non-isolation or isolation of an azol-sensitive species. How many of these antifungal therapies were really active against the isolated species? 

Data on antifungal susceptibility in our patients population are not currently available.

-                     Many factors related with worse outcomes are mainly explicatives since occurring concomitantly to patients evolution: the larger the number of days the worse the outcomes is only reflecting a patient in whom antibiotics are maintained for a longer period since he is doing bad.

We agree with this interpretation and had stated it in the Discussion section (194-196). Outcome analysis was eliminated.

-                     There is a relatively small number of patients with Candida isolation. This is a major limiting factor when trying to perform a retrospective multivariable analysis in such difficult cohort (patients with large admissions and lots of uncontrolled biases and neglected variables). I am not sure if Candida really plays a major pathogen role or is merely  reflexing the disease severity; looking at very low positive rate of B-D-glucan and that all Candida isolation were polymicrobial one would think so...

We agree with you and underly the high limitations of this small cohort in the text. Polymicrobial nature of IAA with Candida isolation has been reported in previous larger studies (Introduction).

-                     It must taken into account that Candida can be difficult to isolate in cultures and maybe we have many false negatives among those patients receiving antifungal therapy. It is logical to think that many of them were negative because of antifugals. In this sense, molecular techniques probably would add t the diagnosis.

We believe that the diagnostics difficulties that you mentioned reflect most real life scenarios, where intra abdominal sample collection is difficult to perform, poor culture outcomes are common especially for fungi and molecular techniques are not available. Despite this, clinicians are compelled to make decisions on antimicrobial therapy even when lacking solid data. For this reason we believe that data from a real life cohort may be of interest.

-                     I suggest authors to provide and assess the prognostic role of other isolated bacteria in the samples analyzed. It would be interesting to see the role of the type of isolated bacteria (BGN, entorococci, polymicrobial).

We agree that ad adverse outcome analysis should include many more variables yet small numbers do not allow for this. We eliminated outcome analysis.

My overall impression after reading the manuscript is that there are many few Candida isolations, there is a large proportion of patients in whom abscesses are not analyzed and there a lot of uncontrolled factors and incoherencies along the manuscript to finally state that empirical antifungal therapy improves outcomes. Manuscript should be completely redone to sound more convincing with these conclusions.

We agree that, based on small numbers, we should focus on reporting our prescription experience stating that we are unable to perform an outcome analysis.

Reviewer 3 Report

To the authors, 

Taddei et al. conducted a retrospective cohort study titled “Empiric Anti-Fungal Therapy for Intra-Abdominal Post-Surgical Abscesses in non-ICU Patients”. The authors sought to to evaluate the factors associated with the choice of giving empiric antifungals to patients with PSAs.

Comments:

1.     Line 81: “Candida spp was isolated in 34 cases” How many cases have been microbiologically tested?

2.     Do you think contamination might have been involved in the 34 candida cases? Please discuss.

3.     Why did you exclude ICU patients?

4.     Table 1: “Antibiotic therapy longer than 14 days (%)”: Was this antibiotic therapy before the identification of candida?

5.     Your data indicate that the use of antifungals was beneficial in those patients who had evident abdominal candida infections. It would also be interesting if the use of antifungals had negative effects in those who had no fungal infection. In other words: Is there a cost of empiric antifungal treatment?

Author Response

Thank you very much for your observations, please find our responses below: 

  1. Line 81: “Candida spp was isolated in 34 cases” How many cases have been microbiologically tested?

We reported in Table 1 that deep specimen collection from the abdomen was “not done” in 143/319 patients, that is around half of our patients and is consistent with real life clinical practice. We obtained a culture positive for Candida spp in 34/176 deep specimens.

  1. Do you think contamination might have been involved in the 34 candida cases? Please discuss.

We mentioned in the Discussion section that contamination, especially by medication gauzes as well as molds could cause false-positive beta-D-glucan blood levels. A risk for contamination is inherently present when environmental microorganism, such as fungi, are isolated in culture. We took account of this potential bias when we decided to include in this study only abdominal specimens freshly-drawn from an abdominal collection (see Methods). To better underline this aspect, we added a sentence in the Discussion section.

  1. Why did you exclude ICU patients?

We decided to include only patients who are cared for by the Infectious Disease Consultation Unit in our Center, expecting more uniformity in antimicrobials prescription. As it is stated in the Materials and Methods section. We aimed at providing our experience of the management of post-surgical abscesses by an ID consultation team, so we believe that introducing patients that are cared for differently (i.e. directly by the Intensive Care Physician and the Hematologist) would bias our population.

  1. Table 1: “Antibiotic therapy longer than 14 days (%)”: Was this antibiotic therapy before the identification of candida?

Duration of therapy takes account of empiric and definitive therapy, since the diagnosis of PSA.

  1. Your data indicate that the use of antifungals was beneficial in those patients who had evident abdominal candida infections. It would also be interesting if the use of antifungals had negative effects in those who had no fungal infection. In other words: Is there a cost of empiric antifungal treatment?

Following revision of patients’ charts, no adverse effects were reported after antifungals administration; this was added to the Results section. An additional cost for antimicrobial therapy, especially echinocandins among antifungals, is foreseeable. However, costs could be properly assessed only after balancing with clinical benefits in a cost-effective analysis, which was beyond the scope of this work.

Round 2

Reviewer 2 Report

I do congratulate authors for their efforts to make a more realistic approach with their population. The study is interesting despite its limitations, showing the uncertatinty regarding Candida infection diagnosis in a real clinical practice scenario. I have only a few minor comments: 

- It must be clarified if analysis of factors associated with Candida isolation refer to the whole population or only to the punctioned (sampled) population. If rerfering to the whole population it is important to sub-analyze it also in the sampled population. We do not know what exactly occurs within non sampled abdomens. 

- When analyzing the type of surgeries I suggest transform ing Uppper/Biliary-pancreas/Lower into one solely category which could be Upper (including HBP) vs. Lower or even an ordinal category with 3 levels: Lower, Upper, HBP clearly seem to have a gradient of risk for Candida presence (OR 0,3; 1,79; 2,25). Analyzing them as if separate categories difficults its statistical association because in fact they are almost the Same category (Upper is not Lower, and Lower is not Upper). In a MV analyisis these complementary approach makes to counteract each other as independent variables. At least one of the variables at UV analysis should be refelected in the MV analysis, unless they are dependent (as this is the case by creating these complementary "categories"). Therefore unifying them into a solely category would overcome these problems related to the few isolations and could add to a practical message: HBP, Upper and Lower surgeries have differents risks for Candida infection.  

- It's a shame that authors do not have data to calculate Candida Score.  

Author Response

We are sincerely grateful for your previous and your additional observations, which we believe substantially  helped to improve our work. We are submitting the text with the required revisions. Please find our responses below:

- It must be clarified if analysis of factors associated with Candida isolation refer to the whole population or only to the punctioned (sampled) population. If rerfering to the whole population it is important to sub-analyze it also in the sampled population. We do not know what exactly occurs within non sampled abdomens. 

  • We appreciate this bias and repeated the analysis on the sub-population with available intra-abdominal samples. Results are shown in Tabel 4 and subsequent text.

- When analyzing the type of surgeries I suggest transform ing Uppper/Biliary-pancreas/Lower into one solely category which could be Upper (including HBP) vs. Lower or even an ordinal category with 3 levels: Lower, Upper, HBP clearly seem to have a gradient of risk for Candida presence (OR 0,3; 1,79; 2,25). Analyzing them as if separate categories difficults its statistical association because in fact they are almost the Same category (Upper is not Lower, and Lower is not Upper). In a MV analyisis these complementary approach makes to counteract each other as independent variables. At least one of the variables at UV analysis should be refelected in the MV analysis, unless they are dependent (as this is the case by creating these complementary "categories"). Therefore unifying them into a solely category would overcome these problems related to the few isolations and could add to a practical message: HBP, Upper and Lower surgeries have differents risks for Candida infection.  

  • Before introducing Lower GI, Upper GI and Epato-biliary surgery in the regression model, dummy variables were created (Lower vs all other, Upper vs all other, etc). We hope this responds to your question.

- It's a shame that authors do not have data to calculate Candida Score.  

  • We agree with you and yet believe that this is the case for most centers, where data such as multiple-site colonization by Candida is not available. We further acknowledged this limitation in the Discussion section.